

# 1 An evaluation of IASI-NH₃ with ground-based FTIR
# 2 measurements

Enrico Dammers[1], Mathias Palm[2], Martin Van Damme[1,3], Corinne Vigouroux[4], Dan Smale[5], Stephanie Conway[6],
Geoffrey C. Toon[7], Nicholas Jones[8], Eric Nussbaumer[9], Thorsten Warneke[2], Christof Petri[2], Lieven Clarisse[3],
Cathy Clerbaux[3], Christian Hermans[4], Erik Lutsch[6], Kim Strong[6], James W. Hannigan[9], Hideaki Nakajima[10],
Isamu Morino[11], Beatriz Herrera[12], Wolfgang Stremme[12], Michel Grutter[12], Martijn Schaap[13], Roy J. Wichink
Kruit[14], Justus Notholt[2], Pierre.-F. Coheur[3] and Jan Willem Erisman[1,15]
1. Cluster Earth and Climate, Department of Earth Sciences, Vrije Universiteit Amsterdam, Amsterdam, the
Netherlands
2. Institut für Umweltphysik, University of Bremen, Bremen, Germany
3. Spectroscopie de l'Atmosphère, Service de Chimie Quantique et Photophysique, Université Libre de Bruxelles
(ULB), Brussels, Belgium
4. Royal Belgian Institute for Space Aeronomy (BIRA-IASB), Brussels, Belgium
5. National Institute of Water and Atmosphere, Lauder, New Zealand
6. University of Toronto, Toronto, Ontario, Canada
7. Jet Propulsion Laboratory, California Institute of Technology, Pasadena
8. University of Wollongong, Wollongong, Australia
9. NCAR, Boulder, Colorado, United States
10. Atmospheric Environment Division, National Institute for Environmental Studies (NIES), Japan
11. National Institute for Environmental Studies, 16-2 Onogawa, Tsukuba, Ibaraki, 305-8506, Japan
12. Centro de Ciencias de la Atmosfera, Universidad Nacional Autonoma de Mexico, Mexico City, Mexico
13. TNO Built Environment and Geosciences, Department of Air Quality and Climate, Utrecht, the Netherlands
14. National Institute for Public Health and the Environment (RIVM), Bilthoven, the Netherlands
15. Louis Bolk Institute, Driebergen, the Netherlands
*Correspondence to*: E. Dammers (e.dammers@vu.nl)
**Abstract.** Global distributions of atmospheric ammonia (NH₃) measured with satellite instruments such as the
Infrared Atmospheric Sounding Interferometer (IASI) contain valuable information on NH₃ concentrations and
variability in regions not yet covered by ground based instruments. Due to their large spatial coverage and daily
observations, the satellite observations have the potential to increase our knowledge of the distribution of NH₃
emissions, and associated seasonal cycles. However the observations remain poorly validated, with only a
handful of available studies often using only surface observations without any vertical information. In this study,
we present the first validation of the IASI-NH₃ product using ground-based Fourier Transform InfraRed (FTIR)
observations. Using a recently developed consistent retrieval strategy, NH₃ concentration profiles have been
retrieved using observations from nine Network for the Detection of Atmospheric Composition Change
(NDACC) stations around the world between 2008- 2015. We demonstrate the importance of strict spatio-
temporal collocation criteria for the comparison. Large differences in the regression results are observed for
changing intervals of spatial criteria, mostly due to terrain characteristics and the short lifetime of NH₃ in the
atmosphere. The seasonal variations of both datasets are consistent for most sites. Correlations are found to be
high at sites in areas with considerable NH₃ levels, whereas correlations are lower at sites with low atmospheric
NH₃ levels close to the detection limit of the IASI instrument. A combination of the observations from all sites
(N_obs = 547) give a MRD of -32.4 ± (56.3) %, a correlation r of 0.8 with a slope of 0.73. These results indicate
that the IASI-NH₃ product performs better than previous upper bound estimates (-50% - +100%).



## 1. Introduction

Humankind has increased the global emissions of reactive nitrogen to an unprecedented level (Holland et al., 1999; Rockström et al., 2009). The current global emissions of reactive nitrogen are estimated to be a factor four larger than pre-industrial levels (Fowler et al., 2013). Consequently atmospheric deposition of reactive nitrogen to ecosystems has substantially increased as well (Rodhe et al., 2002; Dentener et al., 2006). Ammonia ($NH_3$) emissions play a major role in this deposition with a total emission of 49.3Tg in 2008 (Emission Database for Global Atmospheric Research (EDGAR), 2011). Although $NH_3$ emissions are predominantly from agriculture in the Northern Hemisphere, wildfires also play a role, with biomass burning contributing up to 8% of the global emission budget (Sutton et al., 2013). $NH_3$ has been shown to be a major factor in the acidification and eutrophication of soil and water bodies, which threatens biodiversity in vulnerable ecosystems (Bobbink et al., 2010; Erisman et al., 2008, 2011). Through reactions with sulphuric and nitric acid, $NH_3$ also contributes to the formation of particulate matter which is associated with adverse health effects (Pope et al., 2009). Particulate ammonium salts contribute largely to aerosol loads over continental regions (Schaap et al., 2004). Through its role in aerosol formation, $NH_3$ also has an impact on global climate change as hygroscopic ammonium salts are of importance for the aerosol climate effect and thus the global radiance budget (Adams et al., 2001). Furthermore increased $NH_3$ concentrations in the soil also enhance the emission of nitrous oxide ($N_2O$) which is an important greenhouse gas and an ozone-depleting substance (Ravishankara et al., 2009). Finally nitrogen availability is a key factor for the fixation of carbon dioxide ($CO_2$) and thus it is an important factor in climate change.

Despite the fact that $NH_3$ at its current levels is a major threat to the environment and human health, relatively little is known about its total budget and global distribution (Sutton et al., 2013; Erisman et al., 2007). Surface observations are sparse and mainly available for north-western Europe, the United States and China (Van Damme et al., 2015a). At the available sites, in situ measurements are mostly performed with relatively poor temporal resolution due to the high costs of performing reliable $NH_3$ measurements with high temporal resolution. These measurements of $NH_3$ are also hampered by sampling artefacts caused by the reactivity of $NH_3$ and the evaporation of ammonium nitrate (Slanina et al., 2001; von Bobrutzki et al., 2010; Puchalski et al., 2011). As the lifetime of atmospheric $NH_3$ is rather short, on the order of hours to a few days, due to efficient deposition and fast conversion to particulate matter, the existing surface measurements are not sufficient to estimate global emissions without inducing large errors. The lack of vertical profile information further hampers the quantification of the budget, with only a few reported airborne measurements (Nowak et al., 2007, 2010, Leen et al., 2013, Whitburn et al., 2015).

Advanced IR-sounders such as the Infrared Atmospheric Sounding Interferometer (IASI), the Tropospheric Emission Spectrometer (TES), and the Cross-track Infrared Sounder (CrIS) enable retrievals of atmospheric $NH_3$ (Beer et al., 2008; Coheur et al., 2009; Clarisse et al., 2009; Shephard et al., 2011, 2015a). The availability of satellite retrievals provide a means to consistently monitor global $NH_3$ distributions. Global distributions derived from IASI and TES observations have shown high $NH_3$ levels in regions not covered by ground-based data. In this way, more insight was gained into known and unknown $NH_3$ sources worldwide including biomass burning, industry and agricultural areas. Hence, satellite observations have the potential to improve our



knowledge of the distribution of global emissions and their seasonal variation due to their large spatial coverage
and (bi-) daily observations (Zhu et al., 2013; Van Damme et al., 2014b, 2015b; Whitburn et al., 2015; Luo et
al., 2015). However, the satellite observations remain poorly validated with only a few dedicated campaigns
performed with limited spatial, vertical or temporal coverage (Van Damme et al., 2015a; Shephard et al., 2015b,
Sun et al., 2015).
Only a few studies have explored the quality of the IASI-NH$_3$ product. A first evaluation of the IASI
observations was made over Europe using the LOTOS-EUROS model and has shown the respective consistency
of the measurements and simulations (Van Damme et al., 2014b). A first comparison using ground-based and
airborne measurements to validate the IASI-NH$_3$ data set were made in Van Damme et al. (2015a). They
confirmed consistency between the IASI-NH$_3$ data set and the available ground-based observations and showed
promising results for validation by using independent airborne data from the CalNex campaign. Nevertheless,
that study was limited by the availability of independent measurements and suffered from representativeness
issues for the satellite observations when comparing to surface concentration measurements. One of the key
conclusions was the need for vertical profiles (e.g. ground-based remote sensing products or upper-air in situ
measurements to compare similar quantities). Recently, Dammers et al. (2015) developed a retrieval
methodology for Fourier Transform Infrared Spectroscopy (FTIR) instruments to obtain remotely sensed
measurements of NH$_3$ and demonstrated the retrieval characteristics for four sites located in agricultural and
remote areas. Here we explore the use of NH$_3$ total columns obtained with ground based FTIR at nine stations
with a range of NH$_3$ pollution levels to validate the IASI-NH$_3$ satellite product by Van Damme (2014a).
First, we concisely describe the ground based FTIR retrieval and IASI-NH$_3$ product datasets in Sections 2.1 and
2.2. Next we describe the methodology of the comparison in Section 2.3 followed by the presentation of the
results in Section 3, which are then summarized and discussed in Section 4.



## 2. Description of the satellite and FTIR data sets and validation methodology

### 2.1 IASI-NH₃ product

The first global $NH_3$ distribution was obtained by a conventional retrieval method applied to IASI spectra (Clarisse et al., 2009), followed by an in depth case study, using a more sophisticated algorithm, of the sounder's capabilities depending on the thermal contrast (defined in Van Damme et al. (2014a) as the temperature differences between the Earth surface and the atmosphere at 1.5 km altitude, Clarisse et al., 2010). In this study we use the $NH_3$ product developed by Van Damme et al. (2014a). Their product is based on the calculation of a dimensionless spectral index (Hyperspectral Range Index: HRI), which is a quantity representative of the amount of $NH_3$ in the total atmospheric column. This HRI is then converted into $NH_3$ total columns using look-up-tables based on numerous forward simulations for various atmospheric conditions. These look-up-tables relate the HRI and the thermal contrast to a total column of $NH_3$ (Van Damme et al., 2014a). The product includes an error characterization of the retrieved column based on errors in the thermal contrast and HRI. Important advantages of this method over the method by Clarisse (2009) is the relatively small computational cost, the improved detection limit and the ability to identify smaller emission sources and transport patterns above the sea. One of the limitations of this method is the use of only two $NH_3$ vertical profiles: a "source profile" for land cases and a "transported profile" for sea cases (Illustrated in Van Damme et al., 2014a, fig. 3). Another limitation of the product is that it does not allow the calculation of an averaging kernel to account for the vertical sensitivity of the instrument sounding to different layers in the atmosphere. In this paper we will use $NH_3$ total columns retrieved from the IASI-A instrument (aboard of the MetOp-A platform) morning overpass (AM) observations (i.e. 09:30 local time at the equator during overpass) which have a circular footprint of 12 km diameter at nadir and an ellipsoid shaped footprint of up to 20 km x 39 km at the outermost angles. We will use observations from January 1st 2008 to December 31st 2014. Figure 1 shows the mean IASI-NH₃ total column distribution (all observations gridded to a 0.1° x 0.1° grid) using observations above land for the years 2008-2014. The mean columns are obtained through a weighting with the relative error (see Van Damme et al., 2014). The bottom left inset shows the corresponding relative error.





### 2.2  FTIR- NH₃ retrieval

The FTIR-NH$_3$ retrieval methodology used here is described in detail in Dammers et al. (2015) and a summary
is given here. The retrieval is based on the use of two spectral micro-windows, which contain strong individual
NH$_3$ absorption lines. The two spectral windows [930.32-931.32 cm$^{-1}$, MW1] and [962.70-970.00 cm$^{-1}$, MW2]
or the wider version for regions with very low concentrations [929.40-931.40 cm$^{-1}$, MW1 Wide] and [962.10-
970.00 cm$^{-1}$, MW2 Wide] are fitted using SFIT4 (Pougatchev et al., 1995; Hase et al., 2004, 2006) or a similar
retrieval algorithm (Hase et al, 1999) based on the optimal estimation method (Rodgers et al., 2000) to retrieve
the volume mixing ratios (in ppbv) and total columns of NH$_3$ (in molecules cm$^{-2}$). Major interfering species in
these windows include H$_2$O, CO$_2$ and O$_3$. Minor interfering species are N$_2$O, HNO$_3$, CFC-12 and SF$_6$. For the
line spectroscopy, the HITRAN 2012 (Rothman et al., 2013) database is used with a few adjustments for CO$_2$
(ATMOS, Brown et al., 1996), and sets of pseudo-lines generated by NASA-JPL (G.C. Toon) are used for the
broad absorptions by heavy molecules (i.e. CFC-12, SF$_6$). The *a-priori* profiles of NH$_3$ are based on balloon
measurements (Toon et al., 1999) and scaled to fit common surface concentrations at each of the sites. An
exception is made for the a-priori profile at Reunion Island where a modelled profile from the MOZART model
is used (Louisa Emmons, personal communication, 2014). There, the profile peaks at a height of 4-5km, as NH$_3$
are expected to be due to transport of biomass burning emissions on the island and Madagascar. For all stations,
the *a-priori* profiles for interfering species are taken from the Whole Atmosphere Community Climate Model
(WACCM, Chang et al., 2008). Errors in the retrieval are typically ~30% (Dammers et al., 2015), which are
mostly due to uncertainties in the spectroscopy in the line intensities of NH$_3$ and the temperature and pressure
broadening coefficients (HITRAN 2012).

An effort has been made to gather observations from most of the station part of the Network for the Detection of
Atmospheric Composition Change (NDACC) which have obtained relevant solar spectra between 1$^{st}$ of Jan
2008 and 31$^{st}$ of Dec 2014. We excluded stations which have only retrieved or are believed to have, NH$_3$ total
columns smaller than 5x10$^{15}$ (molecules cm$^{-2}$) during the study interval (i.e. Arctic and Antarctic and other
stations with concentrations below the expected limits of the IASI-NH$_3$ product, at best ~5x10$^{15}$ for observations
with high thermal contrast). Figure 1 shows the positions of the FTIR stations used in this study. The retrieved
NH$_3$ total columns (molecules cm$^{-2}$) for each of the stations are shown in Figure 2. The number of available
observations per station varies as does the range in total columns with high values of ~100x10$^{15}$ (molecules cm$^{-2}$) observed at Bremen and low values of about 1x10$^{15}$ (molecules cm$^{-2}$) at St Denis Reunion. The following
provides a short description of each of the sites used in this study and retrieved NH$_3$ columns (molecules cm$^{-2}$).
Additionally, a short summary can be found in Table 1:

The **Bremen** site operated on the university campus by the University of Bremen in the northern part of the city
(Velazco et al., 2007). Bremen is located in the northwest of Germany, which is characterized by intensive
agriculture. It is most suitable for comparisons with IASI given the very high observed concentrations (Fig. 2,
blue) and flat geography surrounding the station. NH$_3$ sources near the measurement station include manure
application to fields, livestock housing and exhaust emissions of local traffic. The retrieved NH$_3$ total columns
peak in spring due to manure application and show an increase in summer due to increased volatilization of NH$_3$
from livestock housing and fields when temperatures increase during summer.





The **Toronto** site (Wiacek et al., 2007) is located on the campus of the University of Toronto, Canada. The city
is next to Lake Ontario with few sources to the south. $NH_3$ sources are mainly due to agriculture as well as local
traffic in the city. Occasionally, $NH_3$ in smoke plumes from major boreal fires to the north and west of the city
can be observed (Lutsch et al., 2016). The retrieved columns (Fig. 2, green) show increased values during
summers as well as peaks in spring.
The **Boulder** observation site is located at the NCAR Foothills Lab in Boulder, Colorado, United States of
America, about 60km northwest of the large metropolitan Denver area. It is located at 1.6 km a.s.l. on the
generally dry Colorado Plateau. Directly to the west are the foothills of the Rocky Mountain range and to the
east are rural grasslands, farming and ranching facilities. Among them are large cattle feed lots to the northeast
near Greeley approximately 90km distant. The area is subject to occasional seasonal local forest fires and also
occasionally sees plumes from fires as distant as Washington or California. The retrieved columns (Fig. 2, grey)
show the largest increase during summers.
The **Tsukuba** site (Ohyama et al., 2009) is located at the National Institute for Environmental Studies (NIES),
in Japan. The region is a mixture of residential and rural zones with mountains to the north. $NH_3$ sources near
the measurement site include manure and fertilizer applications and exhaust emissions of local traffic in the
surrounding city with a large part originating from the from the Tokyo metropolitan area. The retrieved columns
(Fig 2, red) show a general increase during the summers due to increased volatilization rates.
The **Pasadena** site lies on the Northern edge of the Los Angeles conurbation in the United States of America, at
the foot of the San Gabriel mountains which rise steeply to the north to over 1.5 km altitude within 5 km
distance. Local sources of $NH_3$ include traffic, livestock, and occasional fires. FTIR observations typically take
place around local noon to avoid solar obstruction by nearby buildings and morning stratus cloud that is
common May-July. The highest retrieved columns (Fig.2, cyan) are observed during the summers.
The **Mexico City** site is located on the campus of the National Autonomous University of Mexico (UNAM) at
2280 m a.s.l., south of the metropolitan area. Surface $NH_3$ concentrations were measured by active open-path
FTIR during 2003 with typical values between 10 - 40 ppb (Moya et al. 2004). The megacity is host to more
than 22 million inhabitants, over 5 million motor vehicles and a wide variety of industrial activities. Low
ventilation during night and morning causes an effective accumulation of the $NH_3$ and other pollutants in
Mexico City, which is located in a flat basin surrounded by mountains. The concentration and vertical
distribution of pollutants are dominated by the large emissions and the dynamics of the boundary layer which is
on average 1.5 km height during the IASI morning overpass (Stremme et al., 2009, 2013). The retrieved
columns (Fig.2, orange) show an increase during the summers as well as a large daily variation.
The measurement site on the university campus of **St.-Denis** (Senten et al., 2008) is located on the remote
Reunion Island in the Indian Ocean. Observed $NH_3$ columns (Fig. 2, purple) are usually low due to the lack of
major sources nearby the site but increases are observed during the fire season (Sept.-Nov.) with possible fire
plumes originating from Madagascar, as already observed in another study involving short-lived species
(Vigouroux et al., 2009). Local $NH_3$ emissions include fertilizer applied for sugar cane production and local
biomass burning.
The **Wollongong** site is located on the campus of the University of Wollongong. The city of Wollongong is on
the south east coast of Australia with the University only about 2.5 km from the ocean. The measurement site is
also influenced by a 400m escarpment 1 km to the West, and the city of Sydney 60 km to the north.  $NH_3$





sources come mainly from city traffic, as well as seasonal forest fires that can produce locally high amounts of
smoke and subsequent $NH_3$ emissions (Paton-Walsh et al., 2005). The retrieved columns (Fig.2, brown) peak
during the summer season due to the higher temperatures and seasonal forest fires.
The **Lauder** (Morgenstern et al., 2012) National Institute of Water and Atmospheric Research (NIWA) station
in Central Otago, New Zealand, is located in a hilly region with $NH_3$ emissions in the valley surrounding the
station mostly due to livestock grazing and fertilizer application. The observed columns (Fig. 2, black) show a
general increase during summers due to increased volatilization rates.




### 2.3 FTIR and satellite comparison methodology

#### 2.3.1 Co-location & data criteria

NH$_3$ is highly variable in time and space which complicates the comparison between the IASI and FTIR observations. Therefore collocation criteria were developed to investigate and mitigate the effect of the spatial and temporal differences between the FTIR and IASI observations on their correlation. So far, there is no model to describe the representativeness of a site for the region so a simple criterion was initially derived by analyzing the terrain around each site and comparing the correlation of the IASI and FTIR observations for multiple time and spatial differences to find the best correlation. To illustrate the differences between the representativeness of the sites we take the stations at Bremen, Lauder and Wollongong as examples. Around Bremen the terrain is flat with high reported NH$_3$ emissions (Kuenen et al., 2014) in the region surrounding the city. In contrast, Lauder is located in a hilly region with low NH$_3$ emissions mostly due to local livestock grazing and fertilizer application in the surrounding valleys (EDGAR, 2011). Owing to the flat terrain, the region around Bremen should, in principle, have more homogeneous concentrations than Lauder. A more extreme case for geographical inhomogeneity is Wollongong. Wollongong is located at the coast near a 400m escarpment without major nearby NH$_3$ sources. Hence increasing distances between the satellite measurement pixel center and the station may negatively impact the comparison due to the short lifetime of NH$_3$, and the limitation on transport of NH$_3$ to the site by the terrain (i.e. representativeness problems). Because no uniform criterion was found that would enable a good comparison for all stations, multiple criteria with a maximum difference of between 10 km and 50 km will be used to analyze the optimal setting for each of the sites.

**Topography**

Any hill or mountain range located between the satellite pixel and the FTIR station may inhibit transport and decrease their comparability. To account for the topography we only used observations which have at maximum an altitude difference of 300 m between the location of the FTIR and the IASI pixel position. The 300 m criterion was chosen based on tests using the FTIR and satellite observations from Lauder. For the calculation of the height differences we used the Space Shuttle Radar Topography Mission Global product at 3 arc second resolution (SRTMGL3, Farr et al., 2007).

**Temporal variation**

NH$_3$ concentrations can vary considerably during the day, with lifetimes as short as a few hours not being uncommon (Dentener and Crutzen, 1994; Bleeker et al., 2009). The variability of the concentrations mainly arises from the variability in emission strengths as influenced by agricultural practices, meteorological, and atmospheric conditions such as temperature, precipitation, wind speed and direction, the development of the boundary layer (which is important as the IASI satellite observations take place around 9.30 local time and thus the boundary layer has not always been fully established), pollution level, and deposition rates. To minimize the effects of this variability on the comparability of the IASI and FTIR observations, satellite observations with a time difference to FTIR observation of no more than 90 minutes were used.

**Product error**



The error of the IASI-NH$_3$ columns derives from errors on the HRI and the thermal contrast (Van Damme et al.,
2014a). Applying relative error filters of 50, 75 and 100% showed that mostly lower concentrations are removed
from the comparison. Consequently, introducing any criteria based on the associated (relative) error will bias
any comparison with FTIR columns towards the higher IASI total columns. Therefore, we decided not to filter
based on the relative error as it skews the range of NH$_3$ column totals.

**Meteorological factors**
The lowest detectable total column of the retrieval depends on the thermal contrast of the atmosphere (Van
Damme et al., 2014a). For example, the retrieval has a minimum detectable NH$_3$ column of around $5\times10^{15}$
molecules cm$^{-2}$ at a thermal contrast of about 12 Kelvin (K) for columns using the "transported" profile. A
thermal contrast of 12 K is chosen as the threshold to ensure the quality of the IASI observations, which
represents a lapse rate of around 8K/km altitude, near standard atmospheric conditions. We excluded data for
T$_{skin}$ temperatures below 275.15 K to introduce a basic filter for snow cover and conditions with frozen soils.
Finally, only IASI observations with a cloud cover below 10% are used.

The complete list of selection criteria is summarized in Table 2.

**Quality of the FTIR observations**
No filters were applied to maximize the number of observations usable in the comparison. The resolution and
detection limit of the FTIR instruments is usually better than that of the IASI instrument, leading to retrieved
columns with, in principle, less uncertainty. Overall the FTIR retrievals show an error of ~30% or less with the
largest errors due to the spectroscopic parameters (Dammers et al., 2015). While artefacts are possible in the
data we did not investigate for specific artefacts and possible impacts.

**2.3.2     Application of averaging kernels**
When performing a direct comparison between two remote sensing retrievals, one should take into account the
vertical sensitivity and the influence of a-priori profiles of both methods. One method to remove the influence of
the a-priori profile and the vertical sensitivity is the application of the averaging kernels of both retrievals to the
retrieved profiles of both products. The IASI-NH$_3$ HRI-based product scheme however, does not produce
averaging kernels thus it is not possible to account for the vertical sensitivity of the satellite retrieval. The effect
of the lack of the satellite averaging kernel is hard to predict. Nonetheless following the method described in
Rodgers and Connor (2003), the FTIR averaging kernel $\boldsymbol{A}$ is applied to the IASI profile $\boldsymbol{x}_{sat}$ to account for the
effects of the a-priori information and vertical sensitivity of the FTIR retrieval (the assumed profiles, called
"land" and "sea" are described in Van Damme et al., 2014a). The IASI profile is first mapped to the altitude grid
of the FTIR profile by using interpolation, forming $\boldsymbol{x}_{sat}^{mapped}$. Applying Eqn. (**1**), the smoothed IASI profile
$\hat{\boldsymbol{x}}_{sat}$ is calculated indicating what the FTIR would retrieve when observing the satellite profile, which is then
used to compute a total column. This profile can then be compared with the FTIR profile.
$$\hat{\boldsymbol{x}}_{sat} = \boldsymbol{x}_{ftir}^{apriori} + \boldsymbol{A}(\boldsymbol{x}_{sat}^{mapped} - \boldsymbol{x}_{ftir}^{apriori}) \qquad\qquad\qquad (1)$$





After the application of the averaging kernel, for each FTIR observation, all satellite observations meeting the
coincident criteria are averaged into a single mean total column value to be compared with the FTIR value. If
multiple FTIR observations match a single satellite overpass, taking into account the maximum time difference,
the FTIR observations are also averaged into a single mean total column value.

**3.  Results**
**3.1  The influence of spatial differences between observations**

Following the approach of Irie et al. (2012) we will first show the correlation $r$, the slope as well as the mean
relative difference (MRD) and the mean absolute difference (MAD) between satellite (y-axis) and FTIR $NH_3$
total columns (x-axis) for each of the sites, as a function of the maximum allowable spatial difference between
the observations (xdiff). The relative difference (RD) is defined here as,

$$RD = \frac{(IASI\ column - FTIR\ column)\ x\ 100}{FTIR\ column}$$    **(2)**

A maximum relative difference of 200% was used to remove extreme outliers from the data, typically
observations under wintertime conditions. The left side of Figure 3 shows the correlation coefficients (blue
lines) and slope (red lines) for a selection of sites as a function of xdiff using a maximum allowed sampling time
difference of 90 minutes. The right side of Figure 3 shows the MRD and MAD between the satellite and FTIR
observations as a function of xdiff. The numbers on the bottom of each of the subfigures show the number of
observations used in the comparison. The values in bold beside the title of each subplot give the mean
concentrations of the IASI and FTIR observations. The bars indicate the standard deviation of the slope (left
side figures) and the relative and absolute differences (right side figures).

For most stations an increasing xdiff (Figure 3) means a decreasing correlation (blue lines) and a changing slope
(either decreasing or increasing with distance, red lines). This can be explained by the local character and high
variation of $NH_3$ emissions/concentration in combination with the locations of the stations. Moving further away
from a source will then generally decrease the relation between the concentration in the air and the emission
source. The same is true for satellite observations of the air concentrations, which have a large footprint
compared to the local character of a point measurement (FTIR) and the emissions. The steepness of this
decrease (or increase) tells us something about the local variation in $NH_3$ concentrations, which can be large for
sites near heterogeneous emission sources or in cases with low transport/turbulence and thus overall relatively
low mixing.

Overall the highest correlations are seen at the Bremen site, which can partially be explained by the overall high
number of observations with high concentrations (more than 15-20e15 molecules $cm^{-2}$) which generally favours
the correlations. The mean column totals as well as the MRD and MAD do not change much except for the
smallest xdiff criteria. The larger changes for observations within 15 km are probably due to the smaller number
of observations (which follows from the relatively few IASI observations directly above or near the stations).
The results show an underestimation of observed columns by IASI with the "all stations" slopes in between





~0.6-0.8. The stations with a lower mean FTIR column totals, such as Toronto and Boulder (as well as
Pasadena, Mexico City, and Lauder shown in the Appendix Figure A1) show lower correlations with most
having slopes below one. The correlations decreasing with mean column totals point towards the product
detection limits of the IASI-NH$_3$ product. The Toronto site has lower correlation coefficients for the smallest
xdiffs, but this seems to be due to the large drop in number of observations for a xdiff of <15 km. For higher
xdiff criteria the correlations of the Toronto site shows results similar to Bremen. The observations at Boulder
also show large differences when including more observations further away from the station. This can be
explained by the land use surrounding the Boulder site. Immediately west of the measurement site is a mountain
range which together with our elevation filter leads to rejection of the observations to the west. To the northeast
there are some major farming areas surrounding the river banks. Correlations do increase with a decreasing
xdiff, suggesting that IASI is able to resolve the large gradients in the NH$_3$ concentrations near the site.

From the correlation analysis as function of spatial coincidence, we conclude that a xdiff value of 25 km is
recommended to make a fair comparison between IASI-NH$_3$ and FTIR. Any criteria smaller than 15 km greatly
reduces the number of observations and statistics. xdiff beyond 25 km further decrease the correlations for the
combined set. From this point onward a xdiff value of 25km will be used.

**3.2  Comparison of FTIR and IASI NH3 data**
Observations from multiple years are used to show the coincident seasonal variability of the FTIR and IASI-
NH$_3$ products for each of the sites (Figure 4, FTIR: blue, IASI: red). Observations are grouped together into a
typical year as there are insufficient collocated observations to show an inter-annual time series. Note the
different scales on the y-axis. Similar seasonal cycles are clearly observed in both datasets for most stations.
Enhanced concentrations in spring are observed for Bremen and Toronto as well as Boulder due to manure
application. Most of the sites show an increase of NH$_3$ during the summer months which is likely due to the
increased volatilization of NH$_3$ as an effect of higher temperatures. Fire events that were earlier captured by
FTIR at St.-Denis in November, as well as in the IASI data, are not observed in the collocated sets, which is due
to a lack of coincident observations. Furthermore, there is a lack of observations in wintertime for most of the
stations either due to low thermal contrast or due to overcast conditions. Tsukuba has observations above the
detection limit but only one year of infrequent observations which is insufficient to show an entirely clear
seasonal cycle. A similar thing can be said for Pasadena where the number of coincident observations are too
few to make meaningful conclusions about the seasonal cycle. In conclusion, IASI reflects similar pollution
levels and seasonal cycles as deduced from the FTIR observations.

Figure 5 and 6 show a direct comparison of the FTIR and IASI NH$_3$ total columns for each station as well as a
combination of all the observations. Correlations, number of observations and slope are shown in the figures.
The MRD and these statistics are also summarized in Table 3. The comparison shows a variety of results. As
before, of all 9 stations Bremen shows the best correlation with a coefficient of determination of r = 0.83 and a
slope of 0.60. The intercept is not fixed at zero. The stations with overall lower observed totals columns (less
than $10 \times 10^{15}$ molecules cm$^{-2}$) show lower correlations. Stations with intermediate concentrations like Toronto
and Boulder show correlations r = ~0.7-0.8. The figure also shows the relatively low number of high



observations for both the FTIR and IASI values as a result of the relatively few FTIR observations during
events. The few outliers can have a disproportional effect on the slope as most of the lower observations are less
accurate due to the detection limits of the instruments. Overall most stations, except St.-Denis and Boulder and
Mexico City, indicate an underestimation by IASI of the FTIR columns ranging from 10-50%. The mean
relative differences for most stations are negative with most showing values in between -22.5 ± (54.0) % for
Bremen down to a -61.3 ± (78.7) % for St.-Denis. The bias shows some dependence on the total columns with
the underestimation being higher at stations with high mean total columns and lower at stations with low mean
total columns. An exception to this are stations with the lowest mean total columns (i.e. St.-Denis and
Wollongong). The differences at St.-Denis might be explained by the fact that most IASI observations are
positioned above water due to restrictions for terrain height differences. A similar thing can be said for
Wollongong which is situated on the coast with hills directly to the inland. Most observations are on the border
of water and land which might introduce errors in the retrieval. The combination of all observations gives a
MRD of -32.4 ± (56.3) %.

**4.    Discussion and conclusions**

Recent satellite products enable the global monitoring of atmospheric concentrations of $NH_3$. Unfortunately, the
validation of the satellite products of IASI (Van Damme et al., 2014a), TES (Shephard et al., 2011) and CrIS
(Shephard et al., 2015a) is very limited and, so far, only based on sparse in-situ and airborne studies. Dammers et
al. (2015) presented FTIR total column measurements of $NH_3$ at several places around the world and demonstrated
that these data can provide information about the temporal variation of the column concentrations, which are more
suitable for validation than ground-level concentrations. Ground-based remote sensing instruments have a long
history for validation of satellite products.  FTIR observations are already commonly used for the validation of
many satellite products, including carbon monoxide (CO), methane ($CH_4$) and nitrous oxide ($N_2O$) (Wood et al.,
2002; Griesfeller et al., 2006; Dils et al., 2006; Kerzenmacher et al., 2012). Furthermore, MAX-DOAS systems
are used for the validation of retrievals for reactive gases (e.g. Irie et al., 2012), whereas AERONET is widely
used to validate satellite-derived aerosol optical depth (e.g. Schaap et al., 2008). The successful comparison
between FTIR and IASI $NH_3$ column reported here can be seen as a first step in the validation of $NH_3$ satellite
products.

In this study, we collected FTIR measurements from nine locations around the world and followed the retrieval
described by Dammers et al. (2015). The resulting datasets were used to quantify the bias and evaluate the
seasonal variability in the IASI-$NH_3$ product. Furthermore, we assessed the colocation criteria for the satellite
evaluation. Additional selection criteria based on thermal contrast, surface temperature, cloud cover and
elevation differences between observations, were applied to ensure the quality of the IASI-$NH_3$ observations.
The FTIR averaging kernels were applied to the satellite profiles to account for the vertical sensitivity of the
FTIR and the influence of the a-priori profiles.

To optimally compare the satellite product to the FTIR observations it is best to reduce the spatial collocation
criterion to the size of the satellite instrument's footprint and allow for a time difference as short as possible.





These considerations are to reduce effects of transport, chemistry and boundary layer growth but limit the
number of coinciding observations significantly. We have shown that the spatial distance between the IASI
observations and the FTIR measurement site is of importance: the larger the distance in space, the lower the
correlation. When there is no exact match in the position of both observations the variations in the spatial
separation lead to correlation coefficients that can greatly change even when changing the spatial criteria (xdiff)
from 10 to 30 km. Reasons for the changes are the local nature of $NH_3$ emissions, the surrounding terrain
characteristics and their influence on local transport of $NH_3$. The small values for spatial and temporal
coincidence criteria show the importance of $NH_3$ sources near the measurement sites when using these
observations for satellite validation. For the validation of the IASI observations, we used a xdiff of less than 25
km, which still showed high correlations while a large number of observations is retained for comparison.

Overall we see a broad consistency between the IASI and FTIR observations. The seasonal variations of both
datasets look similar for most stations. Increased column values are observed for both IASI and FTIR during
summers as the result of higher temperatures, with some sites showing an increase in concentrations due to
manure application and fertilization events in spring (Bremen, Toronto). In general our comparison shows that
IASI underestimates the $NH_3$ total columns, except for Wollongong. The Wollongong site has persistent low
background columns, i.e. observations with a low HRI, to which IASI is not very sensitive, which results in an
overestimation of the observed columns. Overall, correlations range from r ~ 0.8 for stations characterised by
higher $NH_3$ column totals (with FTIR columns up to 80e15 molecules $cm^{-2}$) to low r ~ 0.4-0.5 correlations for
stations, which only have a few to no FTIR observations above $5x10^{15}$ molecules $cm^{-2}$. Hence, the detection
limit or sensitivity of the IASI instrument largely explain the lower correlation values. The combination of all
sites ($N_{obs} = 547$) give a MRD of -32.4 ± (56.3) %, a correlation r of 0.8 with a slope of 0.73.

In comparison to ground-based in situ systems, the FTIR observations have the big advantage to provide coarse
vertical profiles, from which a column can be derived, which are more similar to what the satellite measures and
therefore more useful for validation. Dedicated $NH_3$ validation datasets are needed that better match the
overpass times of satellite instruments like IASI, TES and CrIS. This could be achieved by the addition of $NH_3$
to the NDACC measurement protocols and matching the overpass time of these satellites over these
measurement stations by using of the right spectral filters for detecting $NH_3$. Furthermore, the low number of
NDACC stations and their locations are not optimal for a dedicated validation of $NH_3$ satellite products.
Although these provide a starting point, the small set of stations does not cover the entire range of climate
conditions, agricultural source types and emission regimes. Hence, our validation results should be seen as
indicative. Additional stations or dedicated field campaigns are needed to improve this situation. New stations
should be placed in regions where emissions and geography are homogenous to ensure that stations are
representative for the footprints of the satellites. For validation of satellite products using FTIR measurements a
monitoring and measurements strategy needs to be developed with a representative mixture of locations in
addition to ground level data. The later can cover the spatial variation and different temporal measurements can
be used. The use of IASI and FTIR observations to study $NH_3$ distributions at ground level requires a
combination of model calculations and observations (e.g. Erisman et al., 2005a; 2005b). Such techniques are
required to provide all the necessary details to describe the high spatial and temporal variations in $NH_3$.






The direct comparison of the IASI and FTIR columns is an addition to earlier efforts by Van Damme et al.
(2015a) to validate IASI column observations with surface in situ and airborne observations. Our results
presented here indicate that the product performs better than the previously upper bound estimate of a factor 2
(i.e. -50 to +100%) as reported in Van Damme et al. (2014a). Although we tried to diminish any effect of
sampling time and position it cannot be ruled out completely that these impacts the comparison statistics as the
number of stations is small. Still the picture arising from the different stations is rather consistent, which hints at
other issues that may explain the observed bias. A number of important issues concerning the retrieval
techniques may explain the observed difference. First, the HRI based retrieval used for IASI is intrinsically
different to the optimal estimation based approach used for the FTIR retrieval. An IASI optimal estimation
retrieval for $NH_3$ called FORLI does exist but is not fully operationally used as it is computationally much
slower than the HRI method. Surprisingly a first comparison between the FORLI and HRI based retrieval (see
figure 9, Van Damme et al., 2014a) shows ~30% lower retrieved columns by the HRI scheme, which is very
close to the systematic difference quantified here. Do note that the results are not be fully comparable as the
reported HRI-FORLI comparison was for a limited dataset and no quality selection criteria were applied. We
recommend to further explore the use of the optimal estimation based IASI-$NH_3$ retrieval in comparison to the
FTIR observations. Second, the IASI and FTIR retrievals incorporate the same line spectroscopy database
(HITRAN 2012; Rothman et al., 2013) which removes a possible error due to different spectroscopy datasets.
The spectroscopy is the largest expected cause of error in the FTIR observations with measurement noise being
the close second for sites with low concentrations. An improvement to the line parameters (i.e. line intensity,
pressure and temperature effects) would greatly benefit both the FTIR and IASI retrievals. Thirdly, the HRI
based scheme uses the difference between spectra with and without the spectral signature of $NH_3$. A plausible
cause for error in this scheme is the influence and correlation of interfering species in the same spectral
channels. $H_2O$ lines occur near most of the $NH_3$ spectral lines and interfere with the $NH_3$ lines at the resolution
of the IASI instrument. Humidity levels vary throughout the year with an increase amount of water vapour in
summer conditions. The HRI based scheme uses a fixed amount of water vapour and varying amounts of water
vapour may interfere with the HRI value attributed fully to the $NH_3$ columns. As there is a seasonality in the
water vapour content of the atmosphere (Wagner et al., 2006), any error attributed to water vapour should show
a seasonality in the difference between the IASI and FTIR observations. A seasonality was, however, not visible
although it may be that the number of coincident observations was too small to recognize it. This again shows
the need for dedicated $NH_3$ validation data (e.a. dedicated FTIR observations). Finally, another possible cause of
error is the lack of a varying $NH_3$ profile and the proxy used for thermal contrast to describe the state of the
atmosphere. The sensitivity of the scheme to the concentrations of $NH_3$ in the boundary layer is described by
using a fixed profile for land and sea observations in combination with a thermal contrast based on two layers
(surface and 1.5km) as it is expected that most of the $NH_3$ occurs in the boundary layer. In reality the $NH_3$
profile is highly dynamic due to a varying boundary layer height and changing emissions as well as temperature
changes (e.g. inversions etc) occurring throughout the planetary boundary layer. Not accounting for this can
introduce an error and future HRI based schemes should focus on estimating the possible effects of using only a
specific profile. The use of multiple $NH_3$-profiles in combination with multiple temperature layers would be a
better approximation of state of the atmosphere, although computationally more expensive. The sharp difference





between the sea and land retrieval introduces strong variability in observations near the coast. Furthermore,
observations that are directly on the transition between water and land can introduce problems due to the
varying emissivity. Similar issues have been reported for aerosol retrievals (e.g. Schaap et al., 2008).

Although the FTIR observations offer some vertical information, studies combining this technique with tower or
airborne observations are needed to further improve knowledge and sensitivity of the FTIR and satellite
observations to the vertical distribution of $NH_3$. Without this knowledge, it is not possible to use the
observations for quantitative emission estimates and modelling purposes as no uncertainty on the new estimate
can be given. Approaches similar to the recent study by Shepherd et al. (2015b) using an airborne instrument,
possibly in combination with an FTIR system focused on the overpass of multiple satellite systems for an
extended period of time should be used to establish the sensitivities and biases of the different retrieval products
available from satellite instruments as well as the bias between the satellite and surface instruments. The use of
IASI and FTIR observations to study $NH_3$ distributions at ground level requires a combination of model
calculations and observations. Such techniques are required to provide all the necessary details to describe the
high spatial and temporal variations in $NH_3$.

**Acknowledgements**
This work is part of the research programme GO/12-36, which is financed by the Netherlands Organisation for
Scientific Research (NWO). The Lauder NIWA FTIR program is funded through the New Zealand
government's core research grant framework from the Ministry of Business, Innovation and employment. We
thank the Lauder FTIR team for their contribution. Acknowledgements are addressed to the Université de La
Réunion and CNRS (LACy-UMR8105 and UMS3365) for their support of the Reunion Island measurements.
The Reunion Island data analysis has mainly been supported by the A3C project (PRODEX Program of the
Belgian Science Policy Office, BELSPO, Brussels). The University of Toronto's NDACC contribution has been
supported by the CAFTON project, funded by the Canadian Space Agency's FAST Program. Measurements
were made at the University of Toronto Atmospheric Observatory (TAO), which has been supported by
CFCAS, ABB Bomem, CFI, CSA, EC, NSERC, ORDCF, PREA, and the University of Toronto. Part of this
research was performed at the Jet Propulsion Laboratory, California Institute of Technology, under contract with
NASA. IASI has been developed and built under the responsibility of the "Centre national d'études spatiales"
(CNES, France). It is flown on-board the Metop satellites as part of the EUMETSAT Polar System. The IASI
L1 data are received through the EUMETCast near real-time data distribution service.
The IASI-related activities in Belgium were funded by Belgian Science Policy Office through the IASI.Flow
Prodex arrangement (2014-2018). PFC, LC and MVD also thank the FRS-FNRS for financial support. L.C. is a
research associate with the Belgian F.R.S-FNRS. C. Clerbaux is grateful to CNES for scientific collaboration
and financial support. The National Center for Atmospheric Research is supported by the National Science
Foundation. The Boulder observation program is supported in part by the Atmospheric Chemistry Observations
& Modeling Division of NCAR. The measurement programme and NDACC site at Wollongong has been
supported by the Australian Research Council for many years, most recent by grant DP110101948 and
LE0668470. The Mexico City site was funded through projects UNAM-DGAPA (109914) and CONACYT
(249374, 239618). A. Bezanilla, J. Baylón and E. Plaza are acknowledged for their participation in the



measurements and analysis. We would like to thank David Griffith, Clare Murphy and Voltaire Velazco at the
School of Chemistry, University of Wollongong, for maintaining FTS instrumentation and conducting FTS
measurements. We are grateful to the many colleagues who have contributed to FTIR data acquisition at the
various sites.



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





**Tables**

**Table 1** FTIR stations used in the analysis. The location, longitude, latitude and altitude are given for each station as well as the instrument used for the measurements. Typical emission sources are mentioned in the station specifics tab. The topography described the typography of the region surrounding the site. N gives the number of observations made during the period of interest. Time period gives the period from which data is used. The last columns describes the used algorithm for the retrieval.

| Station Location | Lon | Lat | Altitude (m.a.s.l.) | Instrument | Station specifics | Topography | Time period | N | Retrieval type |
|---|---|---|---|---|---|---|---|---|---|
| Bremen, Germany | 8.85E | 53.10N | 27 | Bruker 125 HR | City, fertilizers, livestock | Flat | 2008-2015 | 278 | Normal |
| Toronto, Canada | 79.60W | 43.66N | 174 | ABB Bomem DA8 | City, fertilizers, biomass burning | On the edge of lake Ontario | 2008-2015 | 1167 | Normal |
| Boulder, United States | 105.26W | 39.99N | 1634 | Bruker 120 HR | Fertilizers, biomass burning, livestock | Mountain range to the west | 2010-2015 | 440 | Normal |
| Tsukuba, Japan | 140.13E | 36.05N | 31 | Bruker 125 HR | Fertilizers, city | Mostly flat, hills to the north | 2014-2015 | 66 | Normal |
| Pasadena, United States | 118.17W | 34.20N | 460 | MKIV_JPL | City, fertilizers, biomass burning | Mountain range to the east | 2010-2015 | 695 | Normal |
| Mexico City, Mexico | 99.18W | 19.33N | 2260 | Bruker Vertex 80 | City, fires, fertilizers | In between mountain ranges | 2012-2015 | 3980 | Normal |
| St.-Denis, Reunion | 55.5E | 20.90S | 85 | Bruker 120 M | Fertilizers, biomass burning, remote | Volcanic | 2008-2012 | 948 | Wide |
| Wollongong, Australia | 150.88E | 34.41S | 30 | Bruker 125 HR | Fertilizers, biomass burning, low emissions | Coastal, hills to the west | 2008-2015 | 3641 | Wide |
| Lauder, New Zealand | 169.68E | 45.04S | 370 | Bruker 120 HR | Fertilizers, livestock | Hills | 2008-2015 | 1784 | Normal |

**Table 2** Applied data filters to the IASI-NH$_3$ product.

| Filter | Filter Criteria |
|---|---|
| Elevation | \|FTIRstation – IASI_Observation\| < 300 m |
| Thermal Contrast | Thermal contrast >12 K |
| Surface Temperature | T > 275.15 K |
| IASI-NH$_3$ retrieval Error | None |
| Cloud cover fraction | <10% |
| Spatial sampling difference | 50km →10km, Δx=5 km |
| Temporal sampling difference | <90 minutes |





Table 3. Summarized results of the comparison between FTIR-NH$_3$ and IASI-NH$_3$ total columns within the coincidence criteria threshold (xdiff < 25 km, tdiff < 90minutes). **N** is the number of averaged total columns, **MRD** is the Mean Relative Difference (in %), **r** and **slope** are the correlation coefficient and slope of the linear regression.

| Sites | N | MRD in % (rms 1σ) | r | slope |
|---|---|---|---|---|
| Bremen | 53 | -22.5±(54.0) | 0.83 | 0.60 |
| Toronto | 170 | -46.0±(47.0) | 0.79 | 0.84 |
| Boulder | 38 | -38.2±(43.5) | 0.76 | 1.11 |
| Tsukuba | 15 | -28.3±(35.6) | 0.67 | 0.57 |
| Pasadena | 16 | -47.9±(30.1) | 0.59 | 0.83 |
| Mexico | 65 | -30.8±(43.9) | 0.64 | 1.14 |
| St.-Denis | 20 | -61.3±(78.7) | 0.65 | 1.26 |
| Wollongong | 62 | 6.0±(74.3) | 0.47 | 0.92 |
| Lauder | 108 | -29.7±(57.3) | 0.55 | 0.77 |
| **Combined** | **547** | **-32.4±(56.3)** | **0.80** | **0.73** |



**Figures**

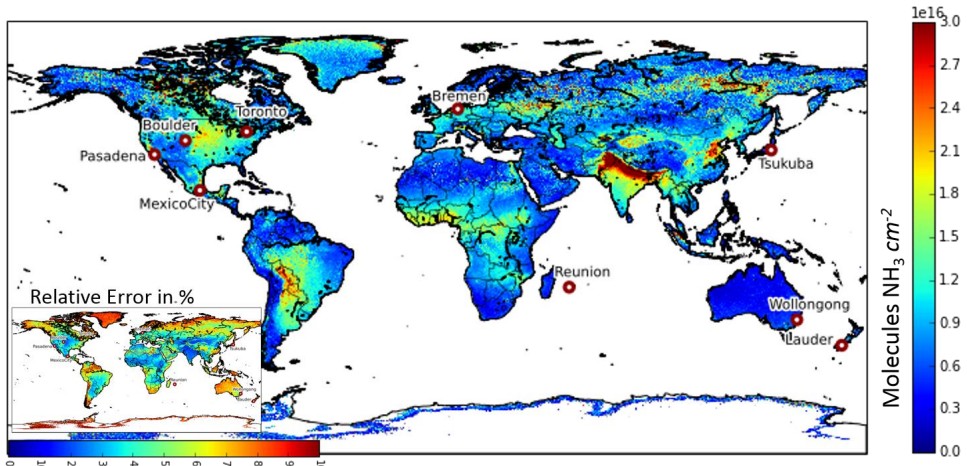

Figure 1. Mean IASI-NH₃ total column distribution for the period between January 2008 and January 2015. The total columns are a weighted average of the individual observations weighted with the relative error. Red circles indicate the positions of the FTIR stations.





Figure 2. FTIR retrieved NH₃ Total Columns (in *molecules cm⁻²*). Note, the labels on the vertical axis vary for each site.



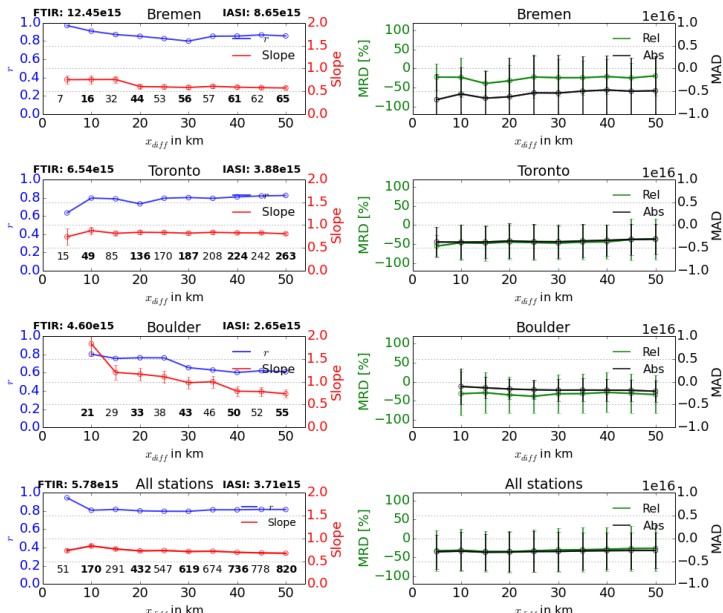

Figure 3. Correlation r (Blue lines, left figures), slope (Red lines, left figures) regression results, Mean Relative Difference (MRD, green lines, right figures) and Mean Absolute Difference (MAD, black lines, right figures) between IASI and FTIR observations as a function of xdiff for a selection of sites. Bars indicate the standard deviation of the slope of the individual regression results. The numbers in the bottom of each subfigure show the number of matching observations. The numbers on the left and right side of the stations name give the mean FTIR and IASI total columns for a xdiff <25km.





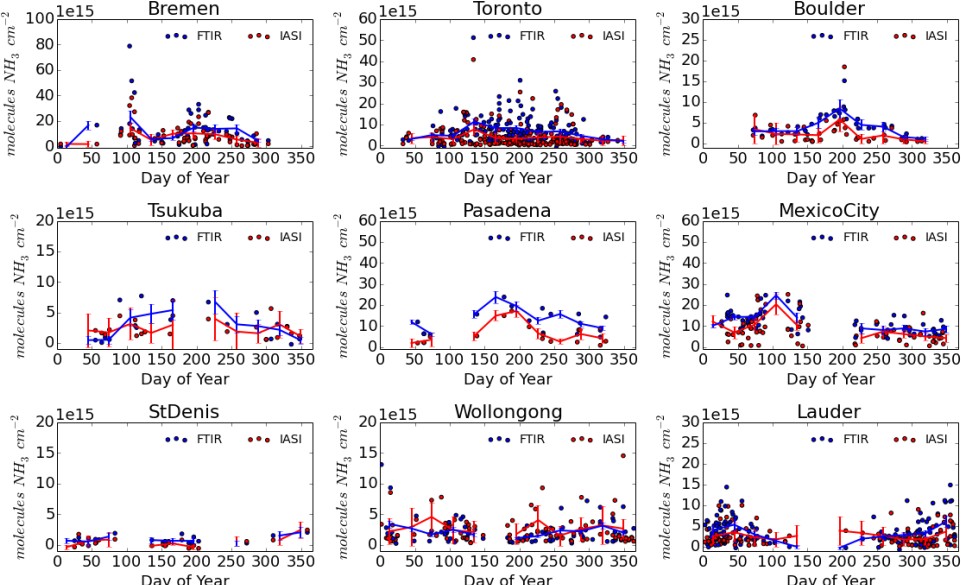

Figure 4. Time series of NH$_3$ for IASI and FTIR datasets with xdiff < 25km and tdiff < 90minutes (FTIR: Blue and IASI: Red). Scattered values are the observations for each day of year (multiple years of observations). The lines show the monthly mean total columns of the respective sets.

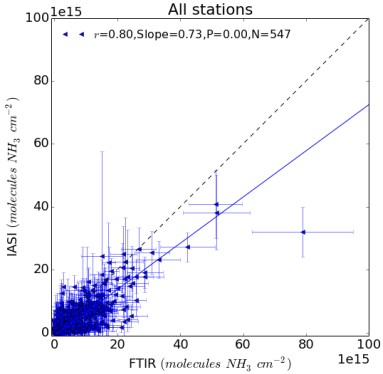

Figure 5. Correlations between the FTIR and IASI total columns with filters thermal contrast > 12K, tdiff < 90min, xdiff < 25km. The trend line shows the results of the regression analysis.





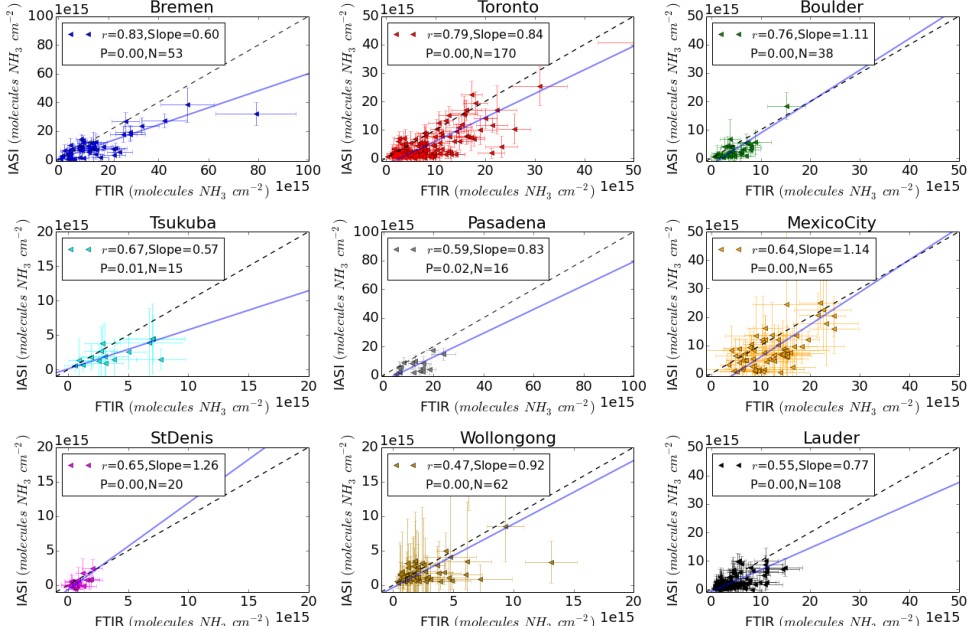

Figure 6. Correlations between the FTIR and IASI total columns with filters thermal contrast > 12, tdiff < 90min, xdiff < 25km. The trend lines show the results of the regression analysis.





## Appendix A

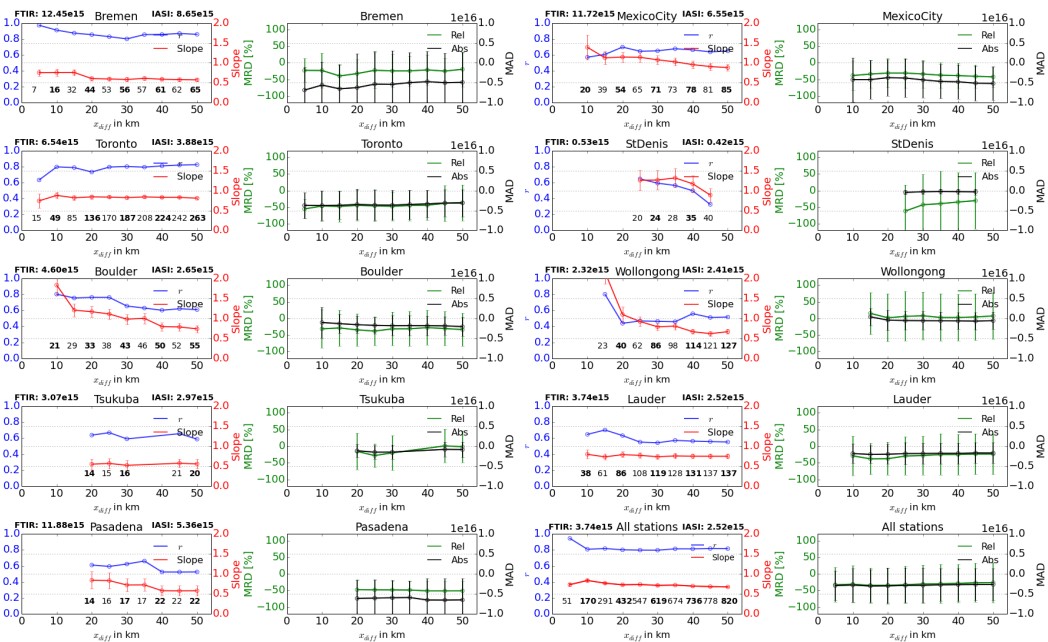

Figure A1. Correlation r (Blue lines, left figures), slope (Red lines, left figures) regression results, Mean Relative Difference (MRD, green lines, right figures) and Mean Absolute Difference (MAD, black lines, right figures) between IASI and FTIR observations as a function of xdiff for all sites. Bars indicate the standard deviation of the slope of the individual regression results. The numbers in the bottom of each subfigure show the number of matching observations. The numbers on the left and right side of the stations name give the mean FTIR and IASI total columns for a xdiff < 25km.

This section further covers the other stations, in addition to the sites covered by section 3.1.
The results for Mexico City show an overall constant correlation coefficient except for small criteria <20km. The slope also decreases towards values seen at other stations. This effect could be due to a large number of sources inside the city, i.e. automobile and agricultural emissions in and near the city, increasing the heterogeneity of the found column totals. Reunion and Tsukuba have few coincident observations leading to only a few significant comparisons. This, combined with the low concentrations measured at Reunion leads to large differences in the mean and standard deviations of the subsequent xdiff sets. The Reunion and Wollongong observations are at the sensitivity limit of the IASI-NH$_3$ retrieval which limits the usefulness of the sites for the validation. As there are only a few observations for Tsukuba it is hard to make meaningful conclusions for the variability around the site.