# Peer review of "An evaluation of IASI-NH3 with ground-based FTIR"

_Atmospheric Chemistry and Physics, 2016_

## Referee Comment (RC1) · Anonymous Referee #1 · 12 Apr 2016

The manuscript by Dammers et al. describes the result of comparisons between ground-based solar absorption FTIR and space-borne mid-infrared nadir observations of NH3 total column amounts. Due to its importance reaching from air-quality to climate change issues, high-quality measurements of ammonia with global coverage are significant contributions for our understanding of relevant processes and the improvement of models. The current manuscript provides one of the first steps in better understanding the quality of space-borne NH3 datasets on a global scale. Due to its large variability such a comparison is complicated and puts high demands on selection of co-incidence criteria – a point which is well captured by the authors. Beyond the quantitative analysis, various reasons for the (mostly) underestimation of NH3 total column amounts by the satellite sensor are discussed which might lead to optimization of the satellite retrieval algorithms. One slight drawback of the paper is that the IASI retrieval approach

used here for the comparison is fast, and unfortunately, does not deliver all means, like averaging kernels, for more in-depth analysis. Thus, I would have liked to see also comparisons to IASI results from other NH3 retrieval schemes. Still, after tackling some of the issues below, I support publication in ACP.

L42: 'give a MRD of -32.4 $\pm$ (56.3) %, . . .These results indicate that the IASI-NH3 product performs better than previous upper bound estimates (-50% - +100%).'

Really better? But -32.4%-56.3% < -50%.

L160: 'We excluded stations which have only retrieved or are believed to have, NH3 total columns smaller than. . .'

However, those cases are also interesting to check for any overestimation of NH3 columns in the IASI dataset (many of the enhancements seen in Figure 1 in remote areas might be artefacts.)

L246: 'To account for the topography we only used observations which have at maximum an altitude difference of 300 m between the location of the FTIR and the IASI pixel position.'

But this criterion does not allow to exclude all cases where there is a mountain between FTIR and IASI measurement but still FTIR and IASI are at the same altitude. It should be extended also to the 'way' between FTIR and IASI position. Can you exclude such a case?

L253:

Please give the information whether the temporal criterion restricts the comparison dataset to the cases of daytime IASI measurements.

L275:

Please specify the source of the skin temperature together with its uncertainty.

L300:

To apply this method seems a bit strange since the satellite profile retrieval is not vertically resolved at all, but the FTIRs are. One should test how much the results change in case this method is not applied. Further, it should be possible to calculate a typical averaging kernel of the IASI retrievals by theoretical simulations.

L407: 'successful comparison'

It is not clear what 'successful' should mean here. Try to be more specific.

L462ff.: possible explanation for the negative bias of satellite data.

Don't one expect an underestimation of total columns from satellite mid-IR observations especially for gases with maxima very near to the surface due to the small thermal contrast there? The FTIR instruments, however, observe the entire columns. This difference would be included in case correct satellite averaging kernels could be used. This should be discussed more in detail.

Fig. 6 and general:

Both datasets, FTIR and satellite ones, seem to exclude negative values. Is this correct? If yes, how is it achieved (log-retrieval?) and should this not have an effect on the comparison for low column amounts?

Technical:

L30-32:

the term 'observations' appears 4 times, try to reformulate

L180 and throughout the manuscript:

'60km' -> '60 km' blank between unit and number

Table 1 caption: 'The topography described the typography of the region'

Please correct.

---

## Referee Comment (RC2) · Anonymous Referee #2 · 17 May 2016

This is a review evaluation for the paper titled, "An evaluation of IASI-NH3 with ground-based FTIR measurements", by Dammers et al. Given the paucity of NH3 satellite validations this study provides valuable comparisons results. The authors also provided details responses addressing the technical remarks from the initial evaluation. Thus, there are only a few remaining additional minor technical remarks. One overall point that should be stated clearly is that the IASI observation sensitivity is not taken into consideration in these comparisons given the IASI retrieval approach, which limits the information available to explain the differences seen between the IASI and the FTIR.

1) Section 2.3.1: This section talks about the important spatial and temporal differences between the FTIR and IASI, which is very well done. However, due to the IASI retrieval approach the sometimes equally important vertical sampling difference are not taken into consideration. One sentence should be added stating that this difference cannot

be determined due to the IASI retrieval and is thus ignored in this comparison.

2) Line 246 change the "which" to a "that".

3) Section 2.3.2 lines 292-292: In might be more clear to the reader if the following was added to the end of the sentence,"The effect of the lack of the satellite averaging kernel is hard to predict so the satellite vertical sensitivity is not taken into consideration in this comparison.

4) Also, in this section the authors provided a good response in regards to explaining where the x_sat IASI profiles are coming from, however, this information was not explicitly added to the text. It would be good to add in some the response provided: The IASI profiles are not fully retrieved profiles but the fixed shape profiles used as an assumption in the IASI retrieval, see Van Damme et al., 2015. These fixed profiles are used for scaling purposes to be able to account for the FTIR averaging kernel. Van Damme, M., Clarisse, L., Dammers, E., Liu, X., Nowak, J. B., Clerbaux, C., Flechard, C. R., Galy-Lacaux, C., Xu, W., Neuman, J. A., Tang, Y. S., Sutton, M. A., Erisman, J. W., and Coheur, P. F.: Towards validation of ammonia (NH3) measurements from the IASI satellite, Atmos. Meas. Tech., 8, 1575-1591, doi:10.5194/amt-8-1575-2015, 2015.

5) It would be nice to added in the rationale for why total column averaging kernels were not used as discussed in your response. Just a simple statement acknowledging that total column AK could be used, but this should in principle be the same as the procedure used here . . ..

6) Line 509: the reference "Shepherd" should be "Shephard" to match the reference list.

---

## Author Comment (AC1) · 16 Jun 2016

We would like to thank Referee #1 for his/her time, constructive and helpful comments, edits and suggestions.

L42: 'give a MRD of -32.4 $\pm$ (56.3) %, . . .These results indicate that the IASI-NH3 product performs better than previous upper bound estimates (-50% - +100%).' Really better? But -32.4%-56.3% < -50%. The sentence did not entirely reflect the meaning. Former estimates were made on an expert guess basis/ comparison with ground observations. The new estimate is the first which is fully based on column measurements and a better estimate of the performance of the product.

Line 42 changed to: These results give an improved estimate of the IASI-NH3 product performance compared to the previous upper bound estimates (-50% - +100%).

L160: 'We excluded stations which have only retrieved or are believed to have, NH3 total columns smaller than. . .' However, those cases are also interesting to check for any overestimation of NH3 columns in the IASI dataset (many of the enhancements seen in Figure 1 in remote areas might be artefacts.)

We agree on this with the reviewer. However because of time restriction we chose to focus on this set of stations. Also we excluded high altitude stations located in regions with large variations of altitude, i.e. Jungfraujoch/Maido. The remaining possible stations/sites are mostly located in the arctic or Antarctic regions and not of direct interest to this study. All observations shown in Figure 1 were used as input in the comparison.

L246: 'To account for the topography we only used observations which have at maximum an altitude difference of 300 m between the location of the FTIR and the IASI pixel position.' But this criterion does not allow to exclude all cases where there is a mountain between FTIR and IASI measurement but still FTIR and IASI are at the same altitude. It should be extended also to the 'way' between FTIR and IASI position. Can you exclude such a case?

This is already the case, changed line 246 to: To account for the topography we only used observations which have at maximum an altitude difference of 300 m (in) between the location of the FTIR and the IASI pixel position.

L253: Please give the information whether the temporal criterion restricts the comparison dataset to the cases of daytime IASI measurements.

Only daytime measurements were used in this study, nighttime observations can be compared but the number of coinciding observations is very low due to the small number of nighttime observations (only during summers is the sun still high enough during the late evening ∼local time 21.30). See line nr 128, where it was mentioned that we use the morning overpasses only.

L275: Please specify the source of the skin temperature together with its uncertainty.

Source is the IASI L2 temperature profiles,

Added Line 276: The Tskin temperatures are obtained from the IASI L2 temperature profiles which have an uncertainty of ~2 K at the surface (August et al., 2012).

Added reference: August, T., Klaes, D., Schlüssel, P., Hultberg, T., Crapeau, M., Arriaga, A., O'Carroll, A., Coppens, D., Munro, R. and Calbet, X.: IASI on Metop-A: Operational Level 2 retrievals after five years in orbit, J. Quant. Spectrosc. Radiat. Transf., 113(11), 1340–1371, doi:10.1016/j.jqsrt.2012.02.028, 2012.

L300: To apply this method seems a bit strange since the satellite profile retrieval is not vertically resolved at all, but the FTIRs are. One should test how much the results change in case this method is not applied. Further, it should be possible to calculate a typical averaging kernel of the IASI retrievals by theoretical simulations. The effects are minor for most sites except for the stations with a large number of the IASI "sea" profile retrieved observations, i.e. for Wollongong and St. Denis. Typical averaging kernel; a typical averaging could be calculated, but the discussion remains to be about what is to be "typical". Something more applicable would be multiple "typical" AVK cases depending on terrain/climate classes. Either way this would introduce more uncertainty instead of dealing/solving the current ones.

L407: 'successful comparison' It is not clear what 'successful' should mean here. Try to be more specific. Removed the word "successful"

L462ff.: possible explanation for the negative bias of satellite data. Don't one expect an underestimation of total columns from satellite mid-IR observations especially for gases with maxima very near to the surface due to the small thermal contrast there? The FTIR instruments, however, observe the entire columns. This difference would be included in case correct satellite averaging kernels could be used. This should be discussed more in detail.

This is true, however the exact effect cannot be estimated due to the variability of the

sensitivity from observation to observation. A short section has been added to the discussion; from Line 491 onward:

Fourth, the negative bias of the satellite observations can be expected by the lack of sensitivity to concentrations near the surface. This is of course where the ammonia concentrations usually peak. The FTIR observations however do fully observe the lower layers in the troposphere thus causing a discrepancy. Normally one can correct for this using the averaging kernel of the satellite observations. However, the IASI-NH3 retrieval does not produce an averaging kernel meaning it is not possible to calculate the exact effect. The use of a typical averaging kernel will cause more uncertainty as there is a large day to day variability in the averaging kernels as earlier retrievals showed (Clarisse et al., 2009).

Fig. 6 and general: Both datasets, FTIR and satellite ones, seem to exclude negative values. Is this correct? If yes, how is it achieved (log-retrieval?) and should this not have an effect on the comparison for low column amounts? The IASI-NH3 retrieval does not retrieve negative total columns following the current retrieval procedure. In case of the FTIR retrieval it is possible to get negative values but due to the retrieval restrictions/settings/procedure it is uncommon. For the "per" station comparison a selection was made, as described in the manuscript, to only use the positive values, in principle this indeed effects the comparison for low column amounts and something like an outlier trim function would be more valid.

Figure 5. Shifted the x- and y- limits to better show the negative values Figure 6. Added greyed values to show the selected and not selected values.

Technical: L30-32: the term 'observations' appears 4 times, try to reformulate Changed Line 30-32; Line 30: daily observations to (bi-) daily overpasses. Line 31: surface observations to surface measurements.

L180 and throughout the manuscript: '60km' -> '60 km' blank between unit and number Added a blank space to " km " in lines: L151, L182,L185, L359, L498, table 2, caption

figure 3, figure 4, figure 5, figure 6 and figure A1. Table 1 caption: 'The topography described the typography of the region' Please correct. Changed part of Table 1 caption to: The topography describes the geography of the region surrounding the site.
[Figure]

---

## Author Comment (AC2) · 16 Jun 2016

We would like to thank Referee #2 for his/her time, constructive and helpful comments, edits and suggestions.

One overall point that should be stated clearly is that the IASI observation sensitivity is not taken into consideration in these comparisons given the IASI retrieval approach, which limits the information available to explain the differences seen between the IASI and the FTIR.

This point was already shortly mentioned in section 2.3.2. We've added a section in the discussion following a comment of Referee nr 1. In addition, we would like to point out that the IASI retrieval product does come with uncertainty estimates which characterize IASI's sensitivity. These depend on the thermal contrast and total column of NH3. We

decided to use a filter for on TC, which captures the main sensitivity component to prevent introducing any biases.

1) Section 2.3.1: This section talks about the important spatial and temporal differences between the FTIR and IASI, which is very well done. However, due to the IASI retrieval approach the sometimes equally important vertical sampling difference are not taken into consideration. One sentence should be added stating that this difference cannot be determined due to the IASI retrieval and is thus ignored in this comparison.

Line added to the end of the section: Vertical sampling differences are not taken into consideration in this study however the IASI selection criterion on the thermal contrast is conservative and only those measurements for which IASI has a good sensitivity to surface concentrations are selected.

2) Line 246 change the "which" to a "that".

Changed as suggested.

3) Section 2.3.2 lines 292-292: In might be more clear to the reader if the following was added to the end of the sentence,"The effect of the lack of the satellite averaging kernel is hard to predict so the satellite vertical sensitivity is not taken into consideration in this comparison.

We have added: "... so the satellite vertical sensitivity is only taken into account through the selection criterion on the thermal contrast."

4) Also, in this section the authors provided a good response in regards to explaining where the x_sat IASI profiles are coming from, however, this information was not explicitly added to the text. It would be good to add in some the response provided: The IASI profiles are not fully retrieved profiles but the fixed shape profiles used as an assumption in the IASI retrieval, see Van Damme et al., 2015. These fixed profiles are used for scaling purposes to be able to account for the FTIR averaging kernel. Van Damme, M., Clarisse, L., Dammers, E., Liu, X., Nowak, J. B., Clerbaux, C., Flechard,

C. R., Galy-Lacaux, C., Xu, W., Neuman, J. A., Tang, Y. S., Sutton, M. A., Erisman, J. W., and Coheur, P. F.: Towards validation of ammonia (NH3) measurements from the IASI satellite, Atmos. Meas. Tech., 8, 1575-1591, doi:10.5194/amt-8-1575-2015, 2015.

Sentence was added to line 296, The IASI profiles are not fully retrieved profiles but fixed shape profiles used as an assumption in the IASI retrieval, see Van Damme et al., 2015a. These fixed profiles are used for scaling purposes to be able to account for the FTIR averaging kernel.

5) It would be nice to added in the rationale for why total column averaging kernels were not used as discussed in your response. Just a simple statement acknowledging that total column AK could be used, but this should in principle be the same as the procedure used here . . ..

Line 297; added: A total column averaging kernel could be used instead, but in principle is similar to the procedure described here.

6) Line 509: the reference "Shepherd" should be "Shephard" to match the reference list.

Changed Shepherd to Shephard.
* * *